# Simplified Method of Determination of the Sound Speed in Water on the Basis of Temperature Measurements and Salinity Prediction for Shallow Water Bathymetry

Artur Makar 

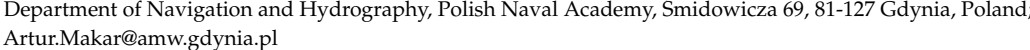

Department of Navigation and Hydrography, Polish Naval Academy, Śmidowicza 69, 81-127 Gdynia, Poland; Artur.Makar@amw.gdynia.pl

**Abstract:** The aim of this paper is to present a method of determining sound speed in water, based on temperature measurements executed by means of a laboratory low-cost thermometer with a probe provided with a long cable. It has been assumed that the salinity variation in respect to depth, found in a shallow water area, has insignificant impact on the sound velocity distribution determined by the temperature changes. The salinity data were obtained via the Internet service from the closest measuring station that registers surface water parameters. The sound speed in water was determined based on the formulas widely adopted in hydroacoustics and compared with the results obtained from the measurements executed by means of a Conductivity/Salinity Temperature Depth (CTD/STD) probe. The impact of inaccuracy in determining the sound speed in respect to the SingleBeam EchoSounder (SBES) immersion depth, i.e., a method commonly used by unmanned surface vessels in seaport measurements, was estimated. The measurements were taken in water areas of the Baltic Sea of low salinity and then verified with measurements in the Mediterranean Sea representing quite high salinity. The method is an alternative for calibrating the SBES the bar check way and has the capacity to meet the requirements in respect to its application in hydrographic surveys.

**Keywords:** sound speed in water; sound velocity profiler; hydrographic surveys; reliability of digital sea bottom model



## 1. Introduction

Determination of the sound speed in water is a basic subject matter in hydrographic measurements. Operation of acoustic devices serving depth measurements is based, very simply stated, on a path equation in a straight-lined linear movement. The distance of the acoustic wave impulse path in water is a route between an electroacoustic transducer and the bottom or another obstacle with reflective qualities. When the sound velocity in water is known, the distance to the bottom is determined based on the measured travelling time along this route. For many years, it was sufficient to know the average speed of sound in the bathymetric measurements executed by means of SBES (SingleBeam EchoSounder). In the depth measurements requiring lower accuracy than in hydrography, e.g., for navigational needs, graphics being lines of constant sound velocity in respect to the temperature used to be applied for the given values of salinity.

It has become necessary to use vertical distribution of the sound speed in water, which is the dependence of the sound velocity on the depth, since MBES (MultiBeam EchoSounder) and underwater navigation systems used to be applied in the bathymetric measurements [1,2]. These devices, contrary to SBES, employ bias propagation of the acoustic wave. As a result of the change of the sound speed in water, its trajectory deflects towards the surface of lower velocity [3–11]. Nonlinear propagation results in more complicated determination of the depth and of the coordinates of the acoustic wave reflection from the bottom. The sound velocity in water profiler becomes an indispensable element of the measuring system.

In general, two types of devices—SVP (Sound Velocity Profiler) and CTD/STD (Conductivity/Salinity Temperature Depth)—are used to measure the sound velocity in water. The first one is an ultrasound device, and its operation principle is similar to the echosounder's one. The velocity is determined on a constant distance between the electroacoustic transducer and a reflective plate, based on the measured time elapsed between the impulse generation and the receipt of its echo after reflection from the plate. The second type—the CTD/STD sounders—measure original water parameters based on which the sound speed is determined. In the 1960s, many teams of hydroacoustic professionals used respective research—the most known relations describing the sound speed in water include Medwin [12,13], Wilson [14], Kinsler and Frey [15], DelGrosso [16], Chen and Millero [17–20], Mackenzie [21,22] and others [23–35].

Application of the sound velocity profiler in MBES systems and in underwater navigation is indisputable. These days, singlebeam echosounders are still in use, in spite of commercial availability of hydrographical multibeam echosounders. Due to emission of one vertical beam, labour intensity in the bathymetric measurements is higher in order to obtain high-density data in the water area under the research. However, they are still in use, especially on USVs (Unmanned Survey Vehicles), due to their small dimensions and uncomplicated calibration process. It is reasonable to look for solutions lowering costs of the sound velocity measurements in order to ensure the highest accuracy of the depth measurement by means of SBES. It may be achieved by determining the sound speed based on the temperature measurement and estimation of the salinity mean value based on the information available from the nearest measuring station [36–44].

Impact of change of the sound speed in water has more and more significance as the depth increases. Inaccuracies between the mean value or estimated distribution and the real distribution of the sound velocity have insignificant impact in shallow water and in restricted water areas of low depths. However, it is obligatory to calibrate SBES in the hydrographic operations and to prove it in the sound velocity measurement sheets. Measurement of the sound speed is the most accurate method of determining its distribution. A bar check calibration is another method. The bar check involves a metal cone or plate device lowered and recording the true depth versus the measured depth and compiling a depth correction table that will be used later to correct the measured depths [45–49].

Generally, the described method is dedicated to hydrographers utilising USV equipped with SBES in shallow waters. The surveys are realized in ports, especially small ones (marinas) with the depth no more 10 m and coastal areas. Deploying a USV in a marina enables manoeuvring between mooring places and yachts. The usage of SBES is sufficient under such circumstances. Similarly, in coastal area, it is easier to survey and manoeuvre a USV closer to the coastline as compared to open water.

Measurements of the sound velocity in water were executed in two water areas: on the Rivers Motława and Martwa Wisła in Gdańsk (A in Figure 1) and in Gdynia Marina (B in Figure 1). The measurements in Gdańsk were executed at the depths of 5 m and 10 m. The first water area serves to station large cruise ships and represents heavy sea traffic in the tourist season. The other one is located in the area of a publicly available slipway, which allowed for the analysis of the accuracy of the depth measurements to be performed in a large range of depths. The measurement station monitoring water parameters is located in the Northern Port in Gdańsk, far from the place the sound velocity measurements were executed. Moreover, it is located on the Gdańsk Bay side.

For the Gdynia area, the sound speed measurements were executed in the marina where the measurement station is located. The bathymetric measurements were carried out at a public beach.

The Baltic Sea belongs to the seas of the lowest salinity—it is often referred to as a subtly salty or brackish sea, not salty. Average salinity of the Baltic Sea is approx. 7 psu—this value usually varies from 2 psu to 12 psu. Salinity values of selected water areas of the Baltic Sea and other seas, including the Mediterranean Sea and salt lakes, are presented in Table 1 [50].

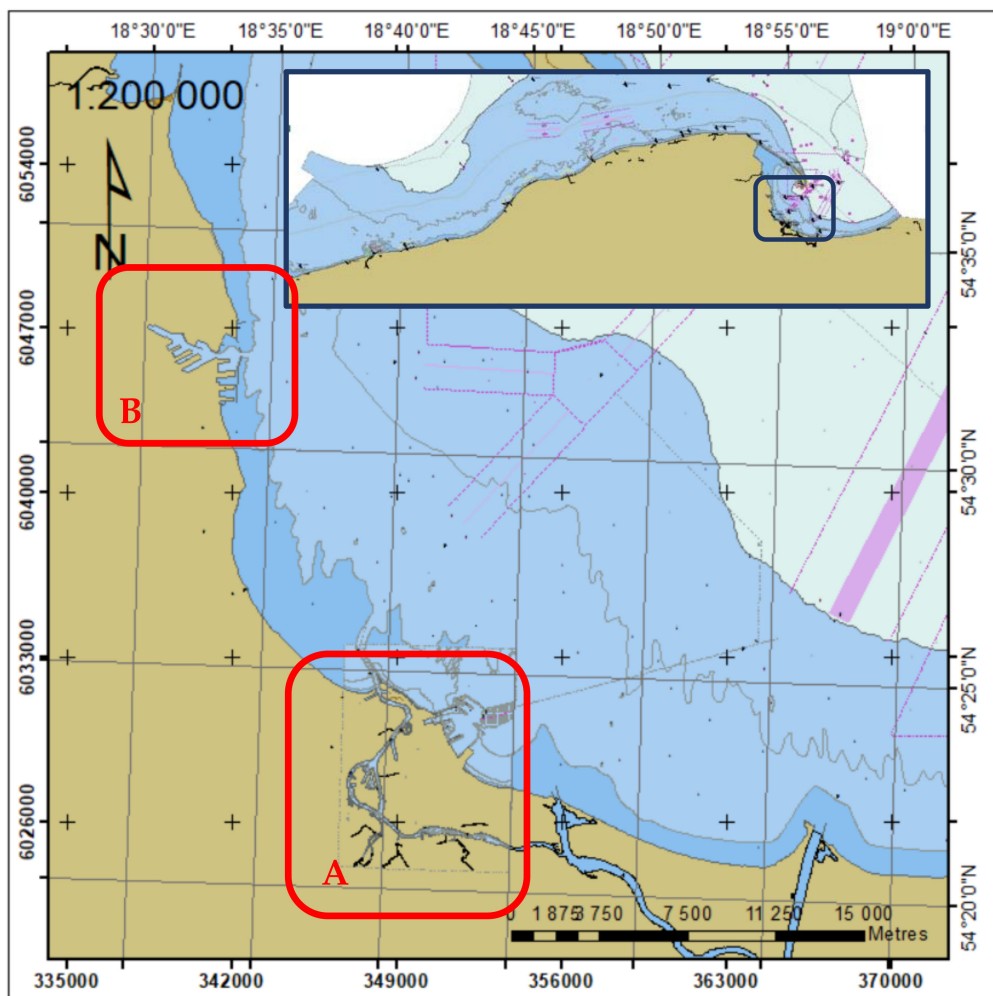

**Figure 1.** Location of sound speed in water and depth measurements: in Gdańsk (A) and Gdynia (B).

**Table 1.** Salinity of seas and lakes.

|  | Area | Salinity [psu] |
|---|---|---|
| Baltic Sea | Kattegat and Skagerrak | 20 |
|  | Bay of Kiel | 15–17 |
|  | Polish coast | 7 |
|  | Bay of Puck | 6.2 |
|  | Vistula Lagoon | 1–3 |
|  | Gulf of Finland and Bothnia | 2 |
| Other seas | Black Sea | 13–23 |
|  | Red Sea | 40–80 |
|  | Mediterranean Sea | 37–39 |
|  | North Sea | 33–35 |
|  | World Ocean | 34–36 |
| Salt lakes | Lake Albert (Great Basin, OR, USA) | 120 |
|  | Dead Sea (Israel, Jordan, West Bank) | 337 |
|  | Gaet'ale Pond (Ethiopia) | 433 |

## 2. Materials and Methods

### 2.1. Determination of the Sound Speed in Water on the Basis of Its Basic Parameters

In physics, the following formula is a basic equation describing velocity of propagation of elastic waves to which sound waves in the medium belong:

$$c = \frac{1}{\sqrt{k_{p,Q}\rho_0}} \tag{1}$$

where: $c$—sound speed (m/s), $k_{p,Q}$—the coefficient of compressibility, $\rho_0$—density and, in practice, it is not applicable in calculating the sound speed in hydrography.

Water density as well as its compressibility are compound functions of the salinity, temperature and pressure [51]. The sound velocity in water increases along with a growth of the temperature, salinity and static pressure. The temperature change has the greatest impact on the sound speed variations. The water compressibility module goes up, and the density of water decreases along with its growth. Change of the sea water temperature by 1 °C results in the sound speed change by the values ranging from 4.7 m/s at the temperature of 0 °C to 2.2 m/s at the temperature of 30 °C. The extent of the sound speed change $\Delta ct$ also depends on the initial temperature of the water in respect to the water in which its change takes place. The water salinity change by 1 psu has impact on the sound speed variation in the range of 1.0–1.4 m/s and the hydrostatic pressure change by 1 atm. (approximately 105 Pa), which means a depth change of 10 m results in the sound velocity change by 0.175 m/s.

Description of the formula c = f(T,S,D) is executed with a use of equations or tables. Medwin elaborated on one of the first and simplest formulas [12–14,52]:

$$\begin{aligned} c(S,T,D) = {} & 1449 + 4.6T - 0.055TT^2 + 0.0003T^3 \\ & + (1.39 - 0.012T)(S - 35) + 0.017D \end{aligned} \tag{2}$$

where: $T$—temperature (°C), $S$—salinity (‰), $D$—depth (m). The other relationships commonly used are the following:

*Wilson* [15]

$$c(S,T,P) = 1449.14 + Dc_T + Dc_S + Dc_P + Dc_{STP}, \tag{3}$$

where: $c_T, c_S, c_P, c_{STP}$—coefficients described in Appendix A.

*Del Grosso* [17]

$$c(S,T,P) = 1402.392 + \Delta c_T + \Delta c_S + \Delta c_P + \Delta c_{STP}. \tag{4}$$

*Mackenzie* [21,22]

$$\begin{aligned} c(S,T,D) = {} & 1448.96 + 4.591T - 5.304{\cdot}10^{-4}T^3 + 1.340(S - 35) + 1.630{\cdot}10^{-2}D \\ & + 1.675{\cdot}10^{-7}D^2 - 1.025{\cdot}10^{-2}T(S - 35) + 7.139{\cdot}10^{-13}TD^3. \end{aligned} \tag{5}$$

*Coppens* [53]

$$\begin{aligned} c(S,t,D) = {} & c(S,t,0) + (16.23 + 0.253t)D + (0.213 - 0.1t)D^2 \\ & + [0.016 + 0.0002(S - 35)](S - 35)tD \\ c(S,t,0) = {} & 1449.05 + 45.7t - 5.21t^2 + 0.23t^3 + \left(1.333 - 0.126t + 0.009t^2\right)(S - 35) \end{aligned} \tag{6}$$

where: $t$ = T/10.

The international standard algorithm, often known as the UNESCO algorithm, is at-tributed to Chen and Millero [19,20] and has a more complicated form than the simple equations above but uses pressure as a variable rather than depth. For the original UNESCO paper, see Fofonoff and Millard [54]. Wong and Zhu [55] recalculated the coefficients in

this algorithm following the adoption of the International Temperature Scale of 1990, and their form of the UNESCO equation is:

$$c(S,T,P) = C_W(T,P) + A(T,P)S + B(T,P)S^{\frac{3}{2}} + D(T,P)S^2. \tag{7}$$

The formula with coefficients is described in Appendix A.

The presented relations serving determination of the sound speed distribution are applicable in specific oceanographic conditions, i.e., in the given ranges of the temperature and salinity. The limits of their usage are presented in Table 2.

**Table 2.** Limits of formulas for determination the sound speed in water.

| Depth | Temperature [°C] | Salinity [psu] |
|---|---|---|
| Medwin | 0–35 | 0–45 |
| Wilson | 0–30 | 0–37 |
| Maccenzie | 2–30 | 25–40 |
| Coppens | 0–35 | 0–45 |
| Del Grosso | 0–30 | 30–40 |
| Chen and Millero | 0–40 | 0–40 |

### 2.2. Observations of Water Parameters

Monitoring and forecasting of environmental changes aim to extend data resources due to the sustainable development needs and in order to prevent risks. Contrary to traditional methods, a digital model controls the main characteristics of an ecosystem constantly, in terms of time and location. Such approach also allows the obtaining of a detailed quantification of variability of physical, dynamic and biochemical water parameters for the marine environment.

The water level is one of the observed parameters. Its significance is important in executing the bathymetric measurements to compare the depth measurement results with a chart datum. At the time of the measurements, the current water level may be obtained from a measurement point, tide gauge, hydrometric station or from GNSS measurements. Permanent monitoring in the observation point is available on the Internet with various intervals (usually 1 h) and age of the data (e.g., 3 days). They constitute national networks or form parts of larger-sized systems [56–58].

The other water parameters, being data sources for engineers in hydrology, oceanography and oceanology, are physical–chemical qualities used, for instance, in eco–hydrodynamic models. It consists of two modules: M3D_UG hydrodynamic one [59] and eco-system ProDeMo module [60,61]. The model works in a preoperational mode. Forty-eight-hour prognoses include fields of surface currents, temperature and salinity of sea water. Moreover, it predicts fields of biogenic salts: nitrates, ammonia, phosphates, silicates, nitrogen and total phosphorus and concentration of oxygen in sea water and phytoplankton biomass. The data from monitoring of the salinity in a surface layer were used to determine the vertical distribution of the sound velocity in water.

### 2.3. Determination of Sound Speed in Water Based on Temperature Measurement and Estimated Salinity

The presented mathematical relations serving to determine the sound speed in water employ the measured values of the temperature and salinity as a function to the depth, which is determined on the basis of hydrostatic pressure. The temperature measurement is easy and inexpensive. For this reason, one may use a laboratory thermometer of limited measuring range, with a probe on a long cable (Figure 2). The measurement is performed by reading the temperature at the depth of the probe immersion, and the depth is recorded by marking the cable with tracers.

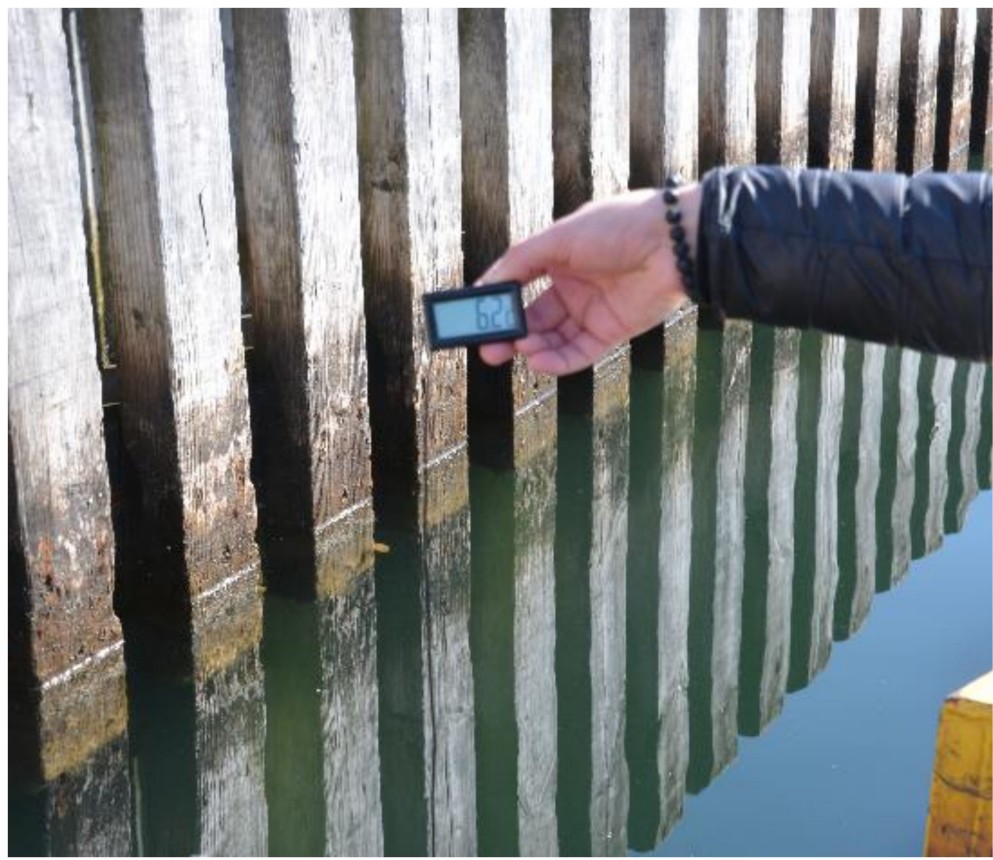

**Figure 2.** Measurement of the water temperature.

The salinity is estimated on the basis of information available in the nearest measurement station or stations registering and providing various parameters of water. The Institute of Oceanology of the University of Gdańsk maintains multiyear registers of: the water temperature, salinity, level, primary production, cyanobacteria, phytoplankton and dissolved oxygen. It also maintains a 48 h prognosis of hydrological and hydrodynamic conditions for the water areas of: the Gdańsk Bay together with the Vistula Lagoon, the Bay of Pomerania and the Szczecin Lagoon as well as the South Baltic.

*2.4. Estimation of the Depth Measurement Accuracy, Utilising the Simplified Method of Sound Velocity in Water Measurement, in the Light of Hydrographical Organisations' Requirements*

Hydrographic organisations have defined minimal requirements for the hydrographic surveys, including an uncertainty of determining the position's coordinates (THU—total horizontal uncertainty) and of the depth measurement (TVU—total vertical uncertainty).

The TPU of a point is a measure of the accuracy to be expected for such a point when all relevant error/uncertainty sources are taken into account. Instead of "TPU", the term "error budget" is also used. Uncertainty sources that are to be considered are (GPS) position, draft, squat, load, tide (including spatial/temporal prediction to the depth measurement position), geoid model, bathy depth, node offsets, timing offsets, SOG (speed over ground) and CMG (course made good), gyro heading, pitch, roll and heave, mounting offsets, beam range, beam angle, beam width, beam steering, sound velocity at transducer head and sound velocity profile.

The accuracy requirements defined in [43,44] are given for five categories, depending on the water area designation. The categories of the highest requirements are as the following:

- Special Order is intended for those areas where underkeel clearances are critical. Therefore, 100% feature search and 100% bathymetric coverage are required. Examples of areas: berthing areas, harbours and critical areas of fairways and shipping channels.
- Exclusive Order hydrographic surveys are an extension of IHO Special Order with more stringent uncertainty and data coverage requirements. Their use is intended to be restricted to shallow water areas (harbours, berthing areas and critical areas of fairways and channels) where there is an exceptional and optimal use of the water column and where specific critical areas with minimum underkeel clearance and bottom characteristics are potentially hazardous to vessels. For this order, a 200% feature search and a 200% bathymetric coverage are required. The size of features to be detected is deliberately more demanding than for Special Order.

The formula below is used to compute the maximum allowable vertical measurement uncertainty [43,44], recognising that there are both depth-dependent and depth-independent error sources that affect the measurements of depths:

$$TVU_{max} = \sqrt{a^2 + (bh)^2}$$ (8)

for: Exclusive Order   a = 0.15  b = 0.0040,
      Special Order     a = 0.25  b = 0.0075.

TVU for Exclusive Order and Special Order in the range of 1–10 m is presented in Table 3.

**Table 3.** TVU for Exclusive Order and Special Order in the range of 1–10 m.

| Depth (m) | 1 | 2 | 3 | 4 | 5 | 6 | 7 | 8 | 9 | 10 |
|---|---|---|---|---|---|---|---|---|---|---|
| Exclusive Order | 0.17 | 0.19 | 0.21 | 0.23 | 0.24 | 0.26 | 0.27 | 0.29 | 0.30 | 0.31 |
| Special Order | 0.26 | 0.28 | 0.29 | 0.30 | 0.32 | 0.33 | 0.34 | 0.35 | 0.36 | 0.37 |

*2.5. STD/CTD Profiler for Sound Speed in Water Measurements*

The water parameters were measured and recorded using SAIV A/S probe model SD204 (Figure 3). The STD/CTD model SD204 is a self-contained instrument that measures, calculates and records sea water conductivity, salinity, temperature, pressure and sound velocity in situ. The data recorded in the instrument are captured in physical units and can be copied to a PC and presented immediately after the measurements have been completed (or at any time later).

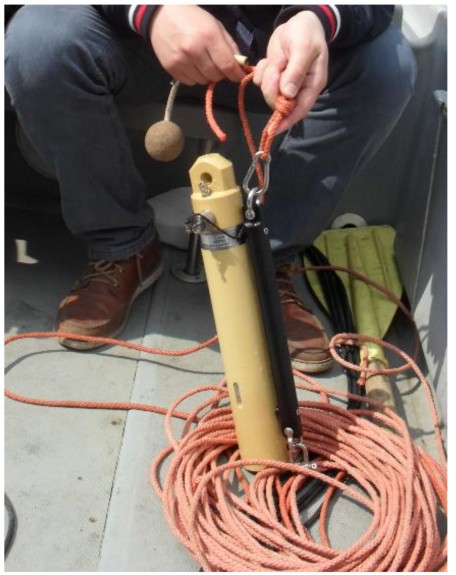

**Figure 3.** STD/CTD probe model SD204.

This makes the performance of the SD204 comparable to much more voluminous and expensive cable-based STD systems. The data from the instrument can also be transferred to a PC by cable or remotely via a modem and a telephone or satellite terminal. The specifications for STD/CTD model SD204 are presented in Table 4.

**Table 4.** Specifications for STD/CTD model SD204.

|  | Range | Resolution | Accuracy |
|---|---|---|---|
| Conductivity | 0 to 70 mS/cm | 0.01 mS/cm | ±0.02 mS/cm |
| Salinity | 0 to 40 ppt | 0.01 ppt | 0.02 ppt |
| Temperature | −2 to +40 °C | 0.001 °C | ±0.01 °C |
| Pressure | 500, 1000, 2000, . . . 6000 m | 0.01 mbar (m) | ±0.02% of range |
| Sound velocity | 1300 to 1700 m/s | 5 cm/s | ±10 cm/s |

## 3. Results

### 3.1. Gdańsk—Motława River

Motława is a river running through the heart of Gdańsk's old town. Although there are insignificant currents in its flow, low depth and serious traffic of vessels in the tourist season have impact on its bed shape, hence the periodic bathymetric control.

Due to a great distance to the measurement station, determination of the sound velocity in water without application of SVP probe is difficult, especially as the station is located in the open water area—in the Gdańsk Bay, in the Northern Port (Figure 4). That is why, in order to determine the distribution of the sound speed in water, its parameters' values—the temperature and salinity—were analysed in distant places, thus taking advantage of the possibility of measuring the salinity in the surface layer.

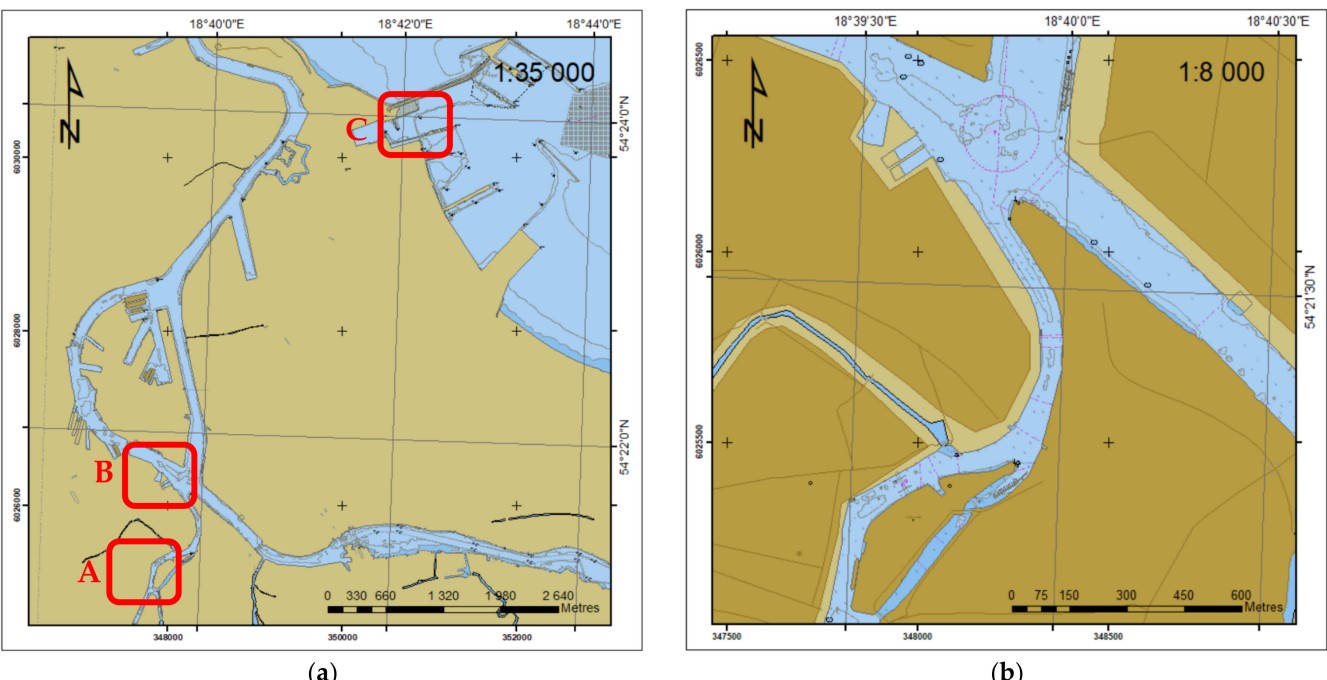

**Figure 4.** Location of measurements of the sound speed in water in Gdańsk (**a**): Motława River (A), Martwa Wisła River (B) and water station in Gdańsk Northern Port (C); Motława River (**b**).

Time courses of water basic parameters: temperature and salinity are presented in Figure 5. A three-day range of changes of these parameters is not high, and it varies in the range of $\Delta t = 1.5$ °C and $\Delta S = 0.5$ psu. A value of $S = 6$ psu has been set to determine the sound velocity in water.

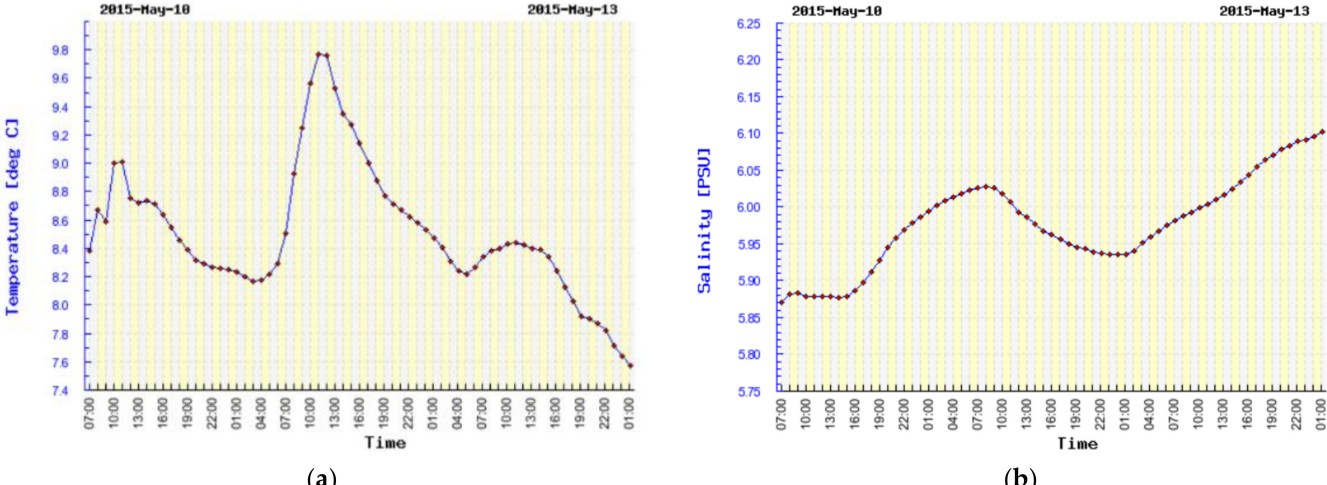

**Figure 5.** Temperature (**a**) and salinity (**b**) recorded in the station in Gdańsk Northern Port in the Motława River for measurements in the Motława River.

The depth reaches 7m along the axis in this part of the Motława River. Shallowing to 5 m occurs in the place of measurement at the water tram pier. Therefore, the sound speed measurements were executed for such a value as a maximum, with an interval of 1m. The sound velocities were determined at those depths for the given, commonly used formulas (Table 5). Then, the sound velocities for the measured temperature values were determined, assuming that the water salinity was S = 6 psu.

**Table 5.** Sound speed in water determined using selected formulas.

| Depth (m) | 0 | 1 | 2 | 3 | 4 | 5 |
|---|---|---|---|---|---|---|
| Temperature (°C) | 14.6 | 13.8 | 13.1 | 12.5 | 11.6 | 10.8 |
| Salinity (psu) | 1 | 2.1 | 3.4 | 4.5 | 5.8 | 6.5 |
| Medwin | 1464.9 | 1463.4 | 1462.5 | 1461.7 | 1459.9 | 1458.0 |
| Wilson | 1465.9 | 1464.3 | 1463.2 | 1462.2 | 1460.4 | 1458.1 |
| Maccenzie | 1464.9 | 1463.4 | 1462.4 | 1461.6 | 1459.8 | 1457.7 |
| Coppens | 1465.6 | 1464.1 | 1463.1 | 1462.3 | 1460.4 | 1458.3 |
| Del Grosso | 1465.7 | 1464.1 | 1463.1 | 1462.2 | 1460.4 | 1458.2 |
| Chen and Millero | 1465.7 | 1464.2 | 1463.2 | 1462.3 | 1460.6 | 1458.4 |
| MTPS [1] (S = 6 psu) | 1471.6 | 1468.8 | 1466.3 | 1464.2 | 1460.8 | 1457.8 |

[1] MTPS—Measured Temperature-Predicted Salinity.

In graphic form, temperature, salinity and sound speed in water profiles in the Motława River are shown in Figure 6.

Based on the registered water parameters, one may observe a change of both temperature and salinity. Along with the depth growth, the temperature decreases within the range of 14.6–10.8 °C and the salinity within the range of 1–6.5 psu. It results in the sound speed change within the range of 1465–1458 m/s. With the assumption that the salinity is 6 psu, the difference between the determined sound velocity in water and its value obtained from the measurements of the temperature and salinity is the highest near the surface, when the salinity variation is the biggest.

In the Motława River, the difference between the determined (MTPS) and measured (on the basis of the temperature and salinity) sound velocity in the water is the highest near the surface. Table 6 shows the depth error as a difference between the depth determined using vertical distribution of the sound speed and on the basis of a simplified method with constant salinity of S = 6 psu. Although the difference in the sound speed is the highest, it has insignificant impact on the depth measurement error.

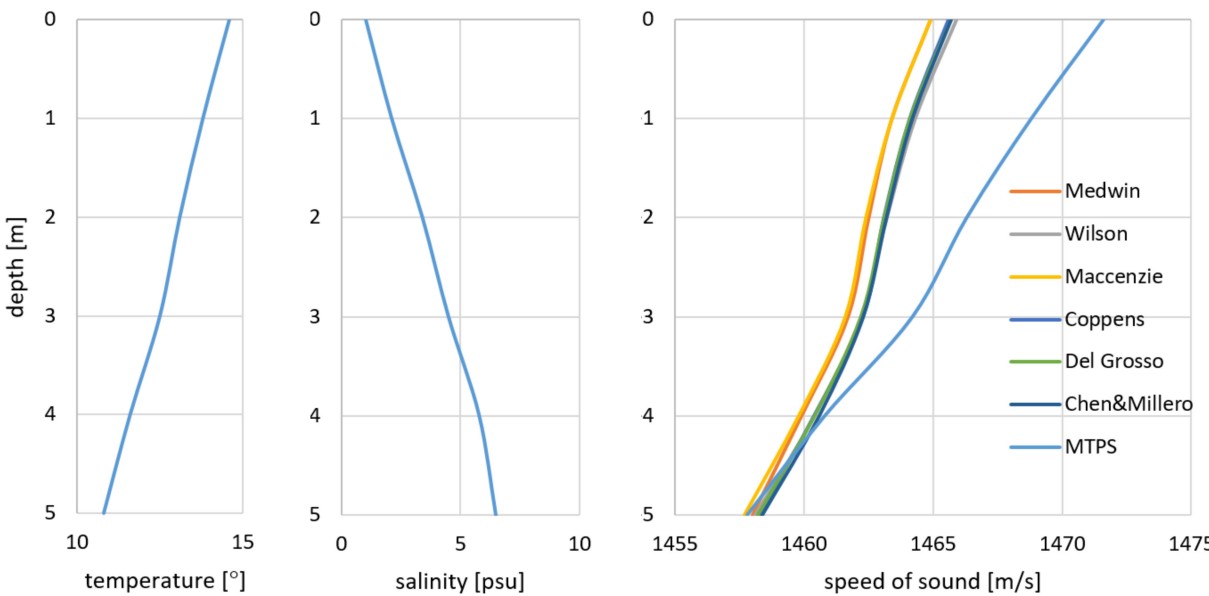

**Figure 6.** Temperature, salinity and sound speed in water profiles in the Motława River.

**Table 6.** Depth measurement error (cm) for selected formulas according to MTPS (S = 6 psu) in the Motława River.

| Depth (m) | 1 | 2 | 3 | 4 | 5 |
|---|---|---|---|---|---|
| Medwin | 0.4 | 0.9 | 1.5 | 2.4 | 3.7 |
| Wilson | 0.3 | 0.8 | 1.4 | 2.3 | 3.7 |
| Maccenzie | 0.4 | 0.9 | 1.5 | 2.5 | 3.8 |
| Coppens | 0.3 | 0.8 | 1.3 | 2.3 | 3.6 |
| Del Grosso | 0.3 | 0.8 | 1.4 | 2.3 | 3.6 |
| Chen and Millero | 0.3 | 0.8 | 1.3 | 2.2 | 3.6 |

*3.2. Gdańsk—Martwa Wisła River*

Bathymetric measurements of the slipway were executed in the area of confluence of the Rivers Motława and Martwa Wisła. It is located near the historic Imperial Shipyard. The registered SBES depth has reached as much as 7 m. The sound speed in water measurement was executed down to the depth of 10 m.

The time courses of the water basic parameters, temperature and salinity, registered by the Gdańsk Northern Port measurement station, are presented in Figure 7.

The extent of changes of these parameters is smaller than for the Motława River. The errors of the depth measurement with the simplified method applied, in respect to the described methods using the salinity measurement, are presented in the Table 7.

**Table 7.** Sound speed in water determined using selected formulas.

| Depth (m) | 0 | 1 | 2 | 3 | 4 | 5 | 6 | 7 | 8 | 9 | 10 |
|---|---|---|---|---|---|---|---|---|---|---|---|
| Temperature (°C) | 9.6 | 7.7 | 8.3 | 8.5 | 8.5 | 8.4 | 8.4 | 8.4 | 8.5 | 8.5 | 8.5 |
| Salinity (psu) | 2.1 | 4.4 | 5.4 | 5.7 | 5.9 | 6.0 | 6.2 | 6.3 | 6.4 | 6.4 | 6.4 |
| Medwin | 1447.6 | 1442.8 | 1446.6 | 1447.8 | 1448.0 | 1447.8 | 1448.0 | 1448.2 | 1448.7 | 1448.7 | 1448.7 |
| Wilson | 1448.1 | 1443.0 | 1446.8 | 1448.0 | 1448.3 | 1448.0 | 1448.2 | 1448.4 | 1449.0 | 1448.9 | 1449.0 |
| Maccenzie | 1447.5 | 1442.7 | 1446.4 | 1447.6 | 1447.9 | 1447.6 | 1447.9 | 1448.0 | 1448.6 | 1448.6 | 1448.6 |
| Coppens | 1448.2 | 1443.7 | 1447.0 | 1448.2 | 1448.5 | 1448.2 | 1448.5 | 1448.6 | 1449.2 | 1449.2 | 1449.2 |
| Del Grosso | 1448.2 | 1443.3 | 1447.0 | 1448.2 | 1448.4 | 1448.1 | 1448.4 | 1448.5 | 1449.0 | 1449.0 | 1449.0 |
| Chen and Millero | 1448.3 | 1443.4 | 1447.2 | 1448.3 | 1448.6 | 1448.3 | 1448.6 | 1448.7 | 1449.2 | 1449.2 | 1449.2 |
| MTPS (S = 6 psu) | 1453.1 | 1445.4 | 1447.9 | 1448.7 | 1448.7 | 1448.3 | 1448.3 | 1448.3 | 1448.7 | 1448.7 | 1448.7 |

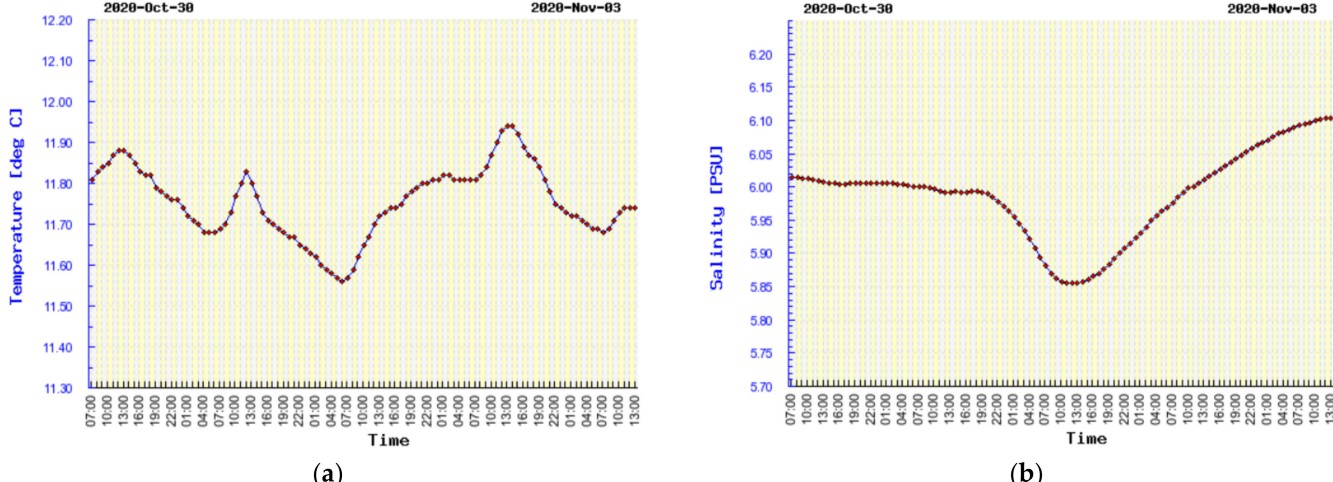

**Figure 7.** Temperature (**a**) and salinity (**b**) recorded in the station in Gdańsk Northern Port for measurements in the Martwa Wisła River.

In graphic form, temperature, salinity and sound speed in water profiles in the Martwa Wisła River are shown in Figure 8.

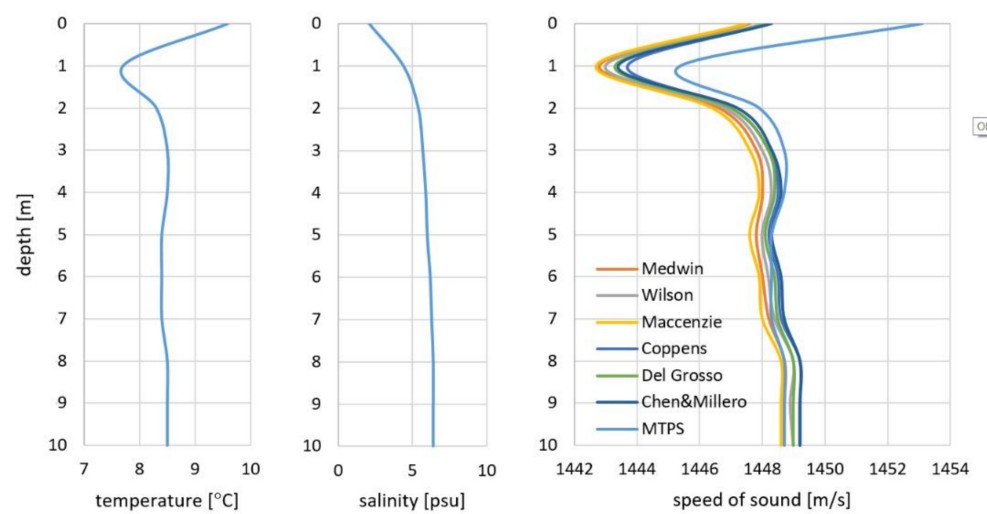

**Figure 8.** Temperature, salinity and sound speed in water profiles in the Martwa Wisła River.

Table 8 shows the depth error as a difference between the depth determined using vertical distribution of the sound speed and on the basis of the simplified method with constant salinity S = 6 psu in the Martwa Wisła River. In the depth range of 0–10 m, the difference is no more than 2.5 cm.

**Table 8.** Depth measurement error (cm) for selected formulas according to MTPS (S = 6 psu) in the Martwa Wisła River.

| Depth (m) | 1 | 2 | 3 | 4 | 5 | 6 | 7 | 8 | 9 | 10 |
|---|---|---|---|---|---|---|---|---|---|---|
| Medwin | 0.2 | 0.2 | 0.5 | 0.7 | 0.8 | 1.1 | 1.4 | 1.8 | 2.1 | 2.3 |
| Wilson | 0.2 | 0.2 | 0.5 | 0.8 | 0.9 | 1.2 | 1.4 | 2.0 | 2.2 | 2.5 |
| Maccenzie | 0.2 | 0.1 | 0.5 | 0.7 | 0.8 | 1.0 | 1.3 | 1.8 | 2.0 | 2.2 |
| Coppens | 0.1 | 0.2 | 0.6 | 0.9 | 1.0 | 1.3 | 1.5 | 2.4 | 2.4 | 2.6 |
| Del Grosso | 0.1 | 0.2 | 0.6 | 0.8 | 0.9 | 1.2 | 1.5 | 2.2 | 2.2 | 2.5 |
| Chen and Millero | 0.1 | 0.2 | 0.6 | 0.9 | 1.0 | 1.3 | 1.6 | 2.4 | 2.4 | 2.6 |

### 3.3. Gdynia—The Marina and the Public Beach

The measurement station in Gdynia is situated in the marina located in the southern part of the basins and piers of the Gdynia seaport (Figure 9). It has direct contact with the Gdańsk Bay and is shielded with two seawalls: one internal and one external.

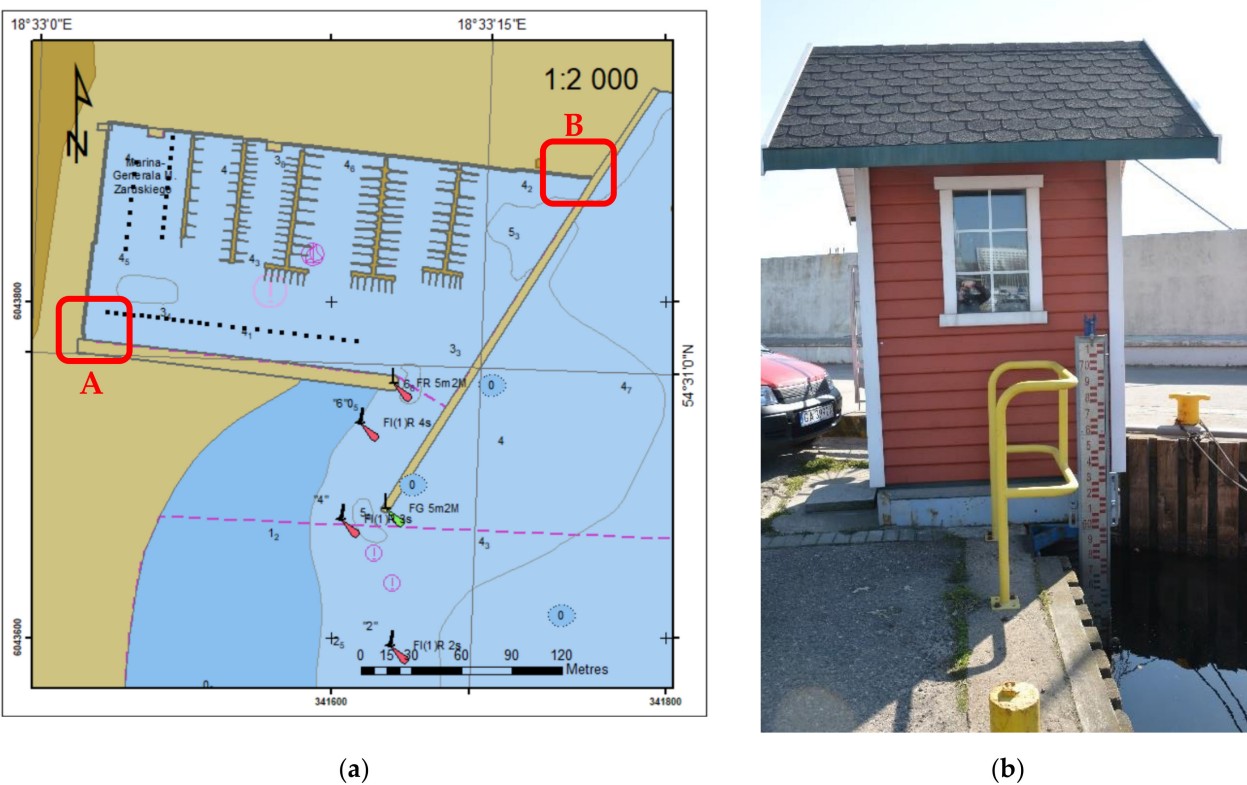

(**a**) (**b**)

**Figure 9.** Location of measurements of the sound speed in water in Gdynia (**a**), A and water station (**b**), B.

Measurements of vertical distribution of the sound speed in water were executed in that place and the bathymetric measurements at the area of the public beach. Similar to the marina, where the measurement of the sound velocity in water were executed, the depth reaches a value of 5 m in that place.

The temperature and salinity recorded in the station in Gdynia for measurements in marina and the public beach are shown in Figure 10.

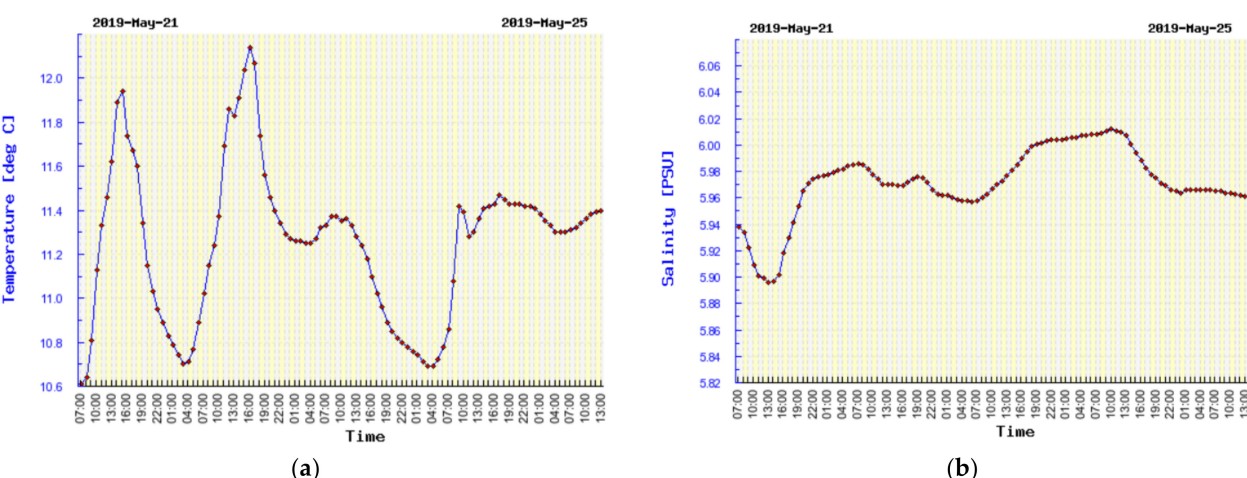

(**a**) (**b**)

**Figure 10.** Temperature (**a**) and salinity (**b**) recorded in the station in Gdynia for measurements in the marina and public beach.

The range of the temperature changes is insignificant, and the salinity is almost constant (Table 9). Thus, the estimated value of the salinity does not result in errors in the sound speed determination.

**Table 9.** Sound speed in water determined using selected formulas.

| Depth (m) | 0 | 1 | 2 | 3 | 4 | 5 |
|---|---|---|---|---|---|---|
| Temperature (°C) | 15.8 | 15.2 | 14.6 | 14.2 | 13.8 | 13.1 |
| Salinity (psu) | 6.9 | 6.9 | 7.0 | 7.0 | 7.1 | 7.1 |
| Medwin | 1476.1 | 1474.1 | 1472.2 | 1470.7 | 1469.5 | 1467.0 |
| Wilson | 1476.7 | 1474.7 | 1472.7 | 1471.3 | 1470.0 | 1467.5 |
| Maccenzie | 1476.1 | 1474.1 | 1472.1 | 1470.7 | 1469.5 | 1467.0 |
| Coppens | 1476.6 | 1474.6 | 1472.7 | 1471.3 | 1470.0 | 1467.5 |
| Del Grosso | 1476.7 | 1474.6 | 1472.7 | 1471.3 | 1470.0 | 1467.6 |
| Chen and Millero | 1476.8 | 1474.8 | 1472.8 | 1471.5 | 1470.2 | 1467.7 |
| MTPS (S = 7 psu) | 1476.9 | 1474.9 | 1472.8 | 1471.5 | 1470.1 | 1467.6 |

In graphic form, temperature, salinity and sound speed in water profiles in Gdynia are shown in Figure 11.

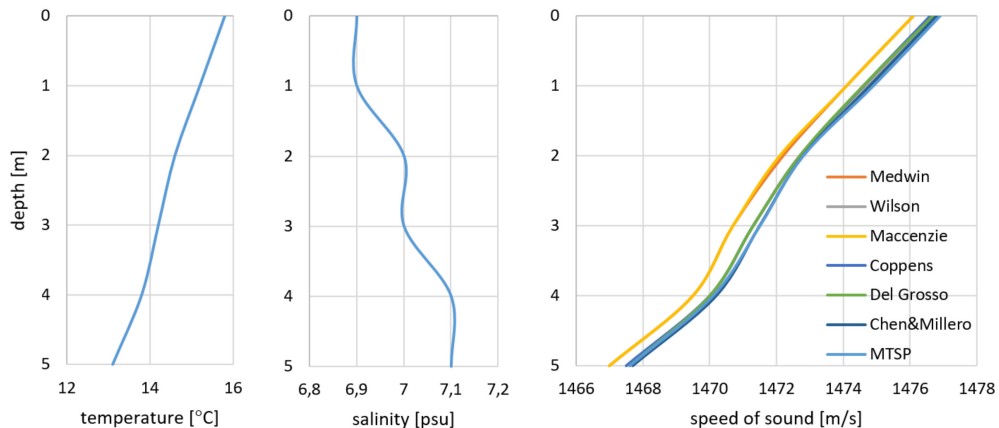

**Figure 11.** Temperature, salinity and sound speed in water profiles in Gdynia.

The depth errors for measurements in a public beach in Gdynia in the range of 0–5 m are presented in Table 10. For constant salinity S = 7 psu, the difference between the depth determined using vertical distribution of the sound speed and on the basis of the simplified method is not more than 3 cm.

**Table 10.** Depth measurement error (cm) for selected formulas according to MTPS (S = 7 psu) in Gdynia.

| Depth (m) | 1 | 2 | 3 | 4 | 5 |
|---|---|---|---|---|---|
| Medwin | 0.1 | 0.4 | 0.9 | 1.5 | 2.7 |
| Wilson | 0 | 0.3 | 0.7 | 1.3 | 2.5 |
| Maccenzie | 0.1 | 0.4 | 0.9 | 1.5 | 2.7 |
| Coppens | 0 | 0.3 | 0.7 | 1.3 | 2.5 |
| Del Grosso | 0 | 0.3 | 0.7 | 1.3 | 2.5 |
| Chen and Millero | 0 | 0.3 | 0.7 | 1.3 | 2.5 |

Depth error for measurements in Gdańsk and Gdynia are shown in Figure 12.

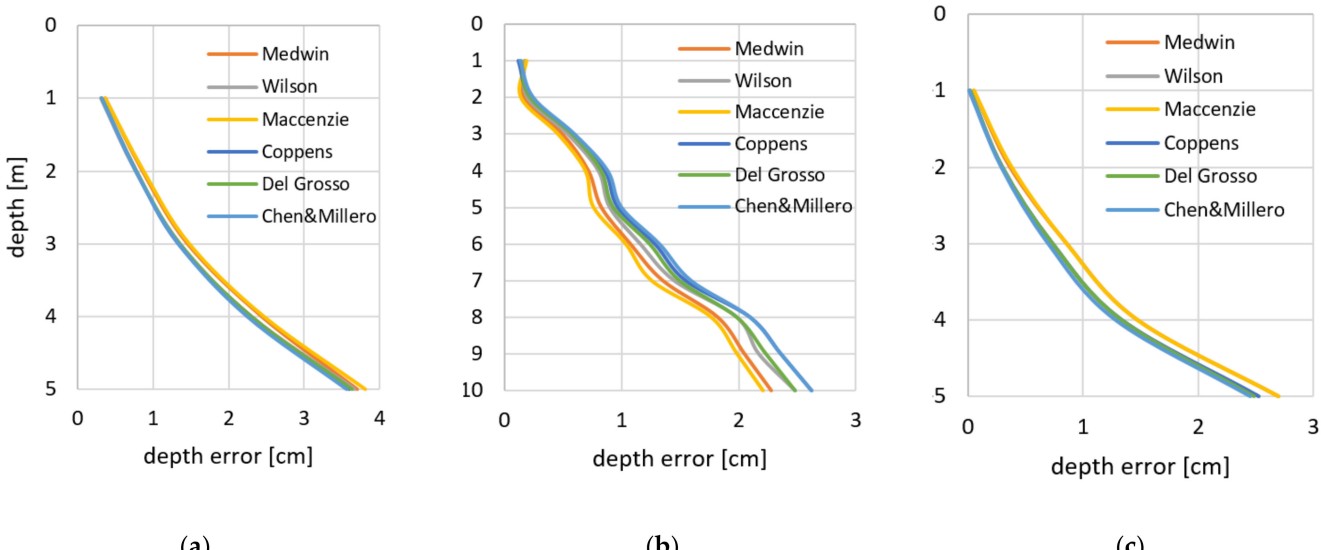

**Figure 12.** Depth error for measurements in the Motława (**a**) and Martwa Wisła (**b**) Rivers and Gdynia (**c**).

### 3.4. Mediterranean Sea—La Ciotat (France) and Barcelona (Spain)

The results given in Section 3.1–3.3 refer to the survey executed in waters of Gdansk Bay, the low-salinity part of the Baltic Sea (Table 1). The results of measurements of water parameters performed in the Mediterranean Sea [62–69] of much higher salinity were used to verify the method for determining the sound velocity distribution.

For validation of the simplified method, the measurements were executed by means of Valeport MIDAS ECM probe in three water areas: La Ciotat (France) 38 psu and Barcelona (Spain) 32 psu and 37 psu. Within the range of 0–15 m of the measured depths, the temperature and the salinity are constant. In such case, the sound velocity may be calculated for any depth, and it is constant in respect to the depth (Figure 13).

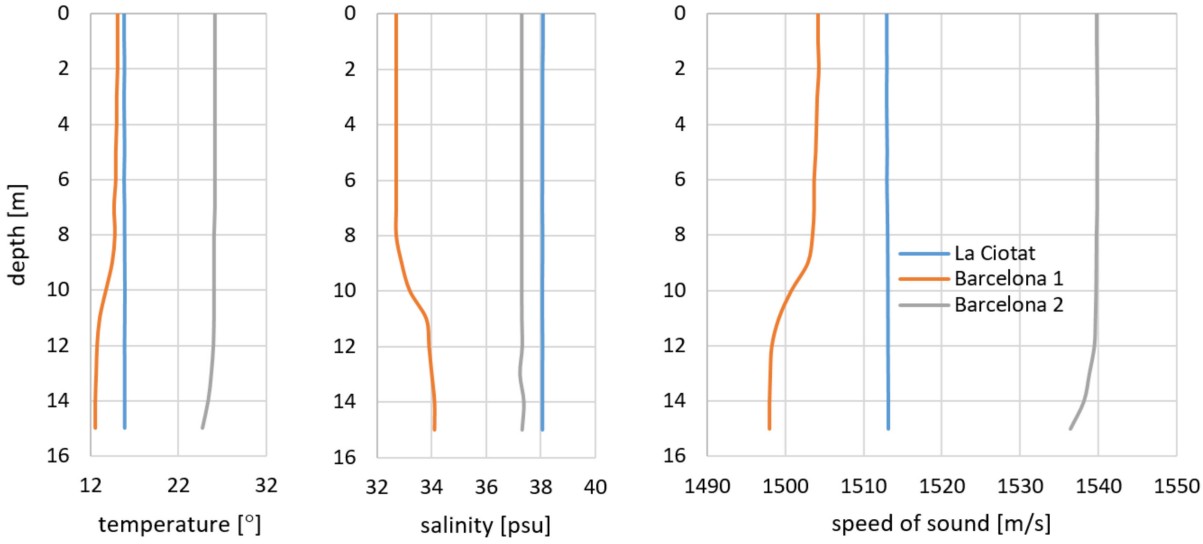

**Figure 13.** Temperature, salinity and sound speed in water profiles in the Mediterranean Sea.

Table 11 presents the results of the measured sound speed in the Mediterranean Sea using Midas SVP probe and determined on the basis of the simplified MTPS method. The difference between the two should be zero (it is 0.1 m/s due to the minimal difference of the salinity assumed for the calculations).

**Table 11.** Sound speed in water determined using selected formulas.

| | Depth (m) | 0 | 2 | 4 | 6 | 8 | 10 | 12 | 14 | 16 |
|---|---|---|---|---|---|---|---|---|---|---|
| **La Ciotat** | Temperature (°C) | 15.8 | 15.8 | 15.8 | 15.8 | 15.8 | 15.8 | 15.8 | 15.8 | 15.8 |
| | Salinity (psu) | 38.0 | 38.0 | 38.0 | 38.0 | 38.0 | 38.0 | 38.0 | 38.0 | 38.0 |
| | Midas SVP (m/s) | 1512.9 | 1512.9 | 1512.9 | 1512.9 | 1513.0 | 1513.1 | 1513.1 | 1513.1 | 1513.1 |
| | MTPS (S = 32 psu) (m/s) | 1512.8 | 1512.8 | 1512.8 | 1512.8 | 1512.9 | 1513.0 | 1513.0 | 1513.0 | 1513.0 |
| **Barcelona 1** | Temperature (°C) | 15.1 | 15.1 | 15.0 | 14.9 | 14.8 | 13.8 | 12.7 | 12,6 | 12.5 |
| | Salinity (psu) | 32.7 | 32.7 | 32.7 | 32.7 | 32.7 | 33.2 | 33.9 | 34.0 | 34.1 |
| | Midas SVP (m/s) | 1504.3 | 1504.3 | 1504.0 | 1503.7 | 153.5 | 1500.8 | 1498.3 | 1498.0 | 1498.0 |
| | MTPS (S = 33 psu) (m/s) | 1504.3 | 1504.3 | 1504.0 | 1503.6 | 1503.4 | 1500.7 | 1498.2 | 1497.9 | 1497.9 |
| **Barcelona 2** | Temperature (°C) | 26.2 | 26.2 | 26.2 | 26.2 | 26.1 | 26.1 | 26.0 | 25.4 | 24.7 |
| | Salinity (psu) | 37.3 | 37.3 | 37.3 | 37.3 | 37.3 | 37.3 | 67.3 | 37.4 | 37.3 |
| | Midas SVP (m/s) | 1539.8 | 1539.8 | 1539.8 | 1539.8 | 1539.8 | 1539.7 | 1537.5 | 1538.1 | 1536.5 |
| | MTPS (S = 37 psu) (m/s) | 1539.7 | 1539.7 | 1539.7 | 1539.7 | 1539.6 | 1539.6 | 1537.4 | 1538.0 | 1536.4 |

## 4. Discussion

The proposed method of determining the sound speed in water on the basis of the temperature measurement and salinity prognosis is dedicated to the measurements in shallow water. In such an area, the bathymetric measurements are executed, more and more often by USV equipped with SBES. Calibration of the echosounder is required, and one the available methods to do that is the bar check. The bar check involves lowering a flat plate below the echo sounder transducer to several known depths below the surface and comparing the actual versus measured depth. As the bar is moved down, the sound velocity in the echo sounder is adjusted until the measured depth matches the actual depth. At the end of the test, the echosounder is fixed with the average sound velocity over the water column. As the sound velocity error magnitude increases proportionally with depth, surveys in shallow water suffer from a smaller potential absolute error. This method is difficult to apply during the measurement executed by the SBES on board of the USV, although it is used in bathymetry, the results of which are presented in literature. The mean value of the sound speed in water is also used, and it causes lower accuracy (higher uncertainty) of the depth measurement. For bathymetric documentation, the sound speed in water or SBES calibration is mandatory. As far as the calibration being awkward for USV's surveys, the simplified MTPS method seems to be a low-cost, easy and effective solution.

The method presented in the article is dedicated to measurements in shallow water because of the length of the probe's cable, usually 1 or 5 m, seldom 10 m long. USVs are usually used in coastal surveys in shallow water, so this low-cost method can be used for supporting hydrographic measurement. During the surveys in open area on board a hydrographic motorboat or vessel, especially in bad weather conditions, measuring the temperature using a light thermometer could be difficult or impossible.

The environment can vary in time and space. In time variation with daily sound speed (temperature) fluctuations, sound speed has to be measured (determined) often. In space variation, e.g., flowing river to the sea, it is necessary to measure the sound of speed in numerous places. Thus, it is possible to model spatial–temporal sound speed in water distribution. In the future, local interpolation for determination local sound speed in water may be realized.

## 5. Conclusions

During hydrographic surveys, the sound speed in water is measured regularly. In daily surveys, it is measured at the beginning and at the end of the survey. In the summer, when the sun heats the water, the measurement period is shorter. When the temperature is measured more often, salinity is rather constant—actual sound speed is determined. Additionally, environmental conditions change in the same way for CTD/STD and simplified MTPS methods.

Singlebeam echosounders determine the depth on the basis of sound speed profile or its mean value. Mean value can be calculated on the basis of sound speed profile or estimated on the basis of constant temperature and salinity. It can also be determined on the basis of constant salinity and actual temperature measured in the transducer or in draught of the transducer. The best solution is to use an SVP probe, but it can be more expensive than the echosounder. The article presents solution on how to obtain accurate sound speed in the water (profile) using low-cost device. The solution is not equally as effective as the SVP probe, but it is better than using the mean value of sound speed and an alternative to bar check calibration.

The presented results of the research have been executed in the limited water area of various seabed shapes. It is important when sound velocity changes along with the depth growth. On one hand, the water salinity and temperature change along with the depth growth resulting in the change of the sound speed. Such an impact may be observed especially in the summertime and to a lesser extent in spring and autumn when the water temperature variations are smaller. On the other hand, in small depths, errors of their measurements are the slightest due to the insignificant change in the sound speed.

Two water areas of various distances to the measurement station were chosen for the research. As far as the first one is concerned (Gdańsk), the sound speed measurement was conducted in water of a river very distant from the station. This may have impact on the reliability of the parameter estimation, i.e., the difference between the values in the measurement place and in the measurement station. The value similar to the real one may be observed during the measurements carried out in Gdynia where the measurement station and the place of the measurements' execution are located in the marina basin.

The application of the proposed method for the hydrological conditions present in the area of South Baltic, of small differences in the salinity in respect to the depth, allows obtaining the accuracy of within several centimetres (but no more than 5cm) as compared to the sound velocity meter.

Positive results can be expected in other areas, where it is possible to estimate the water's salinity. The article presents results obtained in Gdynia, close to the water station, where the salinity spectrum is negligible. The salinity in the surveyed area and at the water station are equal. This is unlike the Motława River, where we can observe variable salinity profile dependent on the measurement's depth.

**Funding:** This research was supported by the Minister of National Defence of Poland as part of the program called Research Grant: "Backscattering of acoustic waves in the aquatic environmental".

**Institutional Review Board Statement:** Not applicable.

**Informed Consent Statement:** Not applicable.

**Data Availability Statement:** Not applicable.

**Conflicts of Interest:** The author declares no conflict of interest.

## Appendix A. Selected Sound Speed in Water Formulas with Coefficients

*Wilson (Equation (3))* [13]

$$c(S, T, P) = 1449.14 + Dc_T + Dc_S + Dc_P + Dc_{STP} \tag{A1}$$

$$Dc_T = 4.5721T - 4.4532 \cdot 10^{-2} T^2 - 2.6045 \cdot 10^{-4} T^3 + 7.9851 \cdot 10^{-6} T^4$$

$$Dc_S = 1.39799(S - 35) - 1.69202 \cdot 10^{-3}(S - 35)^2$$

$$Dc_P = 1.63432P - 1.06768 \cdot 10^{-3} P^2 + 3.73403 \cdot 10^{-6} P^3 - 3.6332 \cdot 10^{-8} P^4$$

$$\begin{aligned} Dc_{STP} = \quad & (S - 35)(-1.1244 \cdot 10^{-2} T + 7.7711 \cdot 10^{-7} T^2 + 7.85344 \cdot 10^{-4} P \\ & -1.3458 \cdot 10^{-5} P^2 + 3.2203 \cdot 10^{-7} PT + 1.3101 \cdot 10^{-8} T^2 P) \\ & +P(-1.8974 \cdot 10^{-3} T + 7.6287 \cdot 10^{-5} T^2 + 4.6176 \cdot 10^{-7} T^3 \\ & +P^2(-2.6301 \cdot 10^{-5} T + 1.9302 \cdot 10^{-7} T^2) + P^3(-2.0831 \cdot 10^{-7} T) \end{aligned}$$

*Del Grosso (Equation (4))* [15]

$$c(S,T,P) = 1402.392 + \Delta c_T + \Delta c_S + \Delta c_P + \Delta c_{STP} \tag{A2}$$

$$\Delta c_T = C_{T1}T + C_{T2}T^2 + C_{T3}T^3$$

$$\Delta c_S = C_{S1}S + C_{S2}S^2$$

$$\Delta c_P = C_{P1}P + C_{P2}P^2 + C_{P3}P^3$$

$$\Delta c_{STP} = \begin{aligned} & C_{TP}TP + C_{T3P}T^3P + C_{TP2}TP^2 + C_{T2P2}T^2P^2 + C_{TP3}TP^3 + C_{ST}ST \\ & + C_{ST2}ST^2 + C_{STP}STP + C_{S2TP}S^2TP + C_{S2P2}S^2P^2 \end{aligned}$$

| Coefficients | Numerical Values |
|---|---|
| $C_{T1}$ | $5.012285$ |
| $C_{T2}$ | $-0.551184 \cdot 10^{-1}$ |
| $C_{T3}$ | $0.221649 \cdot 10^{-3}$ |
| $C_{S1}$ | $1.329530$ |
| $C_{S2}$ | $0.1288598 \cdot 10^{-3}$ |
| $C_{P1}$ | $0.1560592$ |
| $C_{P2}$ | $0.2449993 \cdot 10^{-4}$ |
| $C_{P3}$ | $-0.8833959 \cdot 10^{-8}$ |
| $C_{ST}$ | $-0.1275936 \cdot 10^{-1}$ |
| $C_{TP}$ | $0.6353509 \cdot 10^{-2}$ |
| $C_{T2P2}$ | $0.2656174 \cdot 10^{-7}$ |
| $C_{TP2}$ | $-0.1593895 \cdot 10^{-5}$ |
| $C_{TP3}$ | $0.5222483 \cdot 10^{-9}$ |
| $C_{T3P}$ | $-0.4383615 \cdot 10^{-6}$ |
| $C_{S2P2}$ | $-0.1616745 \cdot 10^{-8}$ |
| $C_{ST2}$ | $0.9688441 \cdot 10^{-4}$ |
| $C_{S2TP}$ | $0.4857614 \cdot 10^{-5}$ |
| $C_{STP}$ | $-0.3406824 \cdot 10^{-3}$ |

*Chen and Millero (Equation (7))* [19,20]

$$c(S,T,P) = C_W(T,P) + A(T,P)S + B(T,P)S^{\frac{3}{2}} + D(T,P)S^2 \tag{A3}$$

$$C_w(T,P) = \begin{aligned} & \left(C_{00} + C_{01}T + C_{02}T^2 + C_{03}T^3 + C_{04}T^4 + C_{05}T^5\right) \\ & + \left(C_{10} + C_{11}T + C_{12}T^2 + C_{13}T^3 + C_{14}T^4\right)P \\ & + \left(C_{20} + C_{21}T + C_{22}T^2 + C_{23}T^3 + C_{24}T^4\right)P^2 \\ & + \left(C_{30} + C_{31}T + C_{32}T^2\right)P^3 \end{aligned}$$

$$A(T,P) = \begin{aligned} & \left(A_{00} + A_{01}T + A_{02}T^2 + A_{03}T^3 + A_{04}T^4\right) \\ & + \left(A_{10} + A_{11}T + A_{12}T^2 + A_{13}T^3 + A_{14}T^4\right)P \\ & + \left(A_{20} + A_{21}T + A_{22}T^2 + A_{23}T^3\right)P^2 \\ & + \left(A_{30} + A_{31}T + A_{32}T^2\right)P^3 \end{aligned}$$

$$B(T,P) = B_{00} + B_{01}T + (B_{10} + B_{11}T)P$$

$$D(T,P) = D_{00} + D_{10}P$$

| Coefficients | Numerical Values | Coefficients | Numerical Values |
|---|---|---|---|
| $C_{00}$ | 1402.388 | $A_{02}$ | $7.166 \cdot 10^{-5}$ |
| $C_{01}$ | 5.03830 | $A_{03}$ | $2.008 \cdot 10^{-6}$ |
| $C_{02}$ | $-5.81090 \ 10^{-2}$ | $A_{04}$ | $-3.21 \cdot 10^{-8}$ |
| $C_{03}$ | $3.3432 \cdot 10^{-4}$ | $A_{10}$ | $9.4742 \cdot 10^{-5}$ |
| $C_{04}$ | $-1.47797 \cdot 10^{-6}$ | $A_{11}$ | $-1.2583 \cdot 10^{-5}$ |
| $C_{05}$ | $3.1419 \cdot 10^{-9}$ | $A_{12}$ | $-6.4928 \cdot 10^{-8}$ |
| $C_{10}$ | 0.153563 | $A_{13}$ | $1.0515 \cdot 10^{-8}$ |
| $C_{11}$ | $6.8999 \cdot 10^{-4}$ | $A_{14}$ | $-2.0142 \cdot 10^{-10}$ |
| $C_{12}$ | $-8.1829 \cdot 10^{-6}$ | $A_{20}$ | $-3.9064 \cdot 10^{-7}$ |
| $C_{13}$ | $1.3632 \cdot 10^{-7}$ | $A_{21}$ | $9.1061 \cdot 10^{-9}$ |
| $C_{14}$ | $-6.1260 \cdot 10^{-10}$ | $A_{22}$ | $-1.6009 \cdot 10^{-10}$ |
| $C_{20}$ | $3.1260 \cdot 10^{-5}$ | $A_{23}$ | $7.994 \cdot 10^{-12}$ |
| $C_{21}$ | $-1.7111 \cdot 10^{-6}$ | $A_{30}$ | $1.100 \cdot 10^{-10}$ |
| $C_{22}$ | $2.5986 \cdot 10^{-8}$ | $A_{31}$ | $6.651 \cdot 10^{-12}$ |
| $C_{23}$ | $-2.5353 \cdot 10^{-10}$ | $A_{32}$ | $-3.391 \cdot 10^{-13}$ |
| $C_{24}$ | $1.0415 \cdot 10^{-12}$ | $B_{00}$ | $-1.922 \cdot 10^{-2}$ |
| $C_{30}$ | $-9.7729 \cdot 10^{-9}$ | $B_{01}$ | $-4.42 \cdot 10^{-5}$ |
| $C_{31}$ | $3.8513 \cdot 10^{-10}$ | $B_{10}$ | $7.3637 \cdot 10^{-5}$ |
| $C_{32}$ | $-2.3654 \cdot 10^{-12}$ | $B_{11}$ | $1.7950 \cdot 10^{-7}$ |
| $A_{00}$ | 1.389 | $D_{00}$ | $1.727 \cdot 10^{-3}$ |
| $A_{01}$ | $-1.262 \cdot 10^{-2}$ | $D_{10}$ | $-7.9836 \cdot 10^{-6}$ |

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
