# Peer review of "Simplified Method of Determination of the Sound Speed in Water on the Basis of Temperature Measurements and Salinity Prediction for Shallow Water Bathymetry"

_remotesensing, doi:10.3390/rs14030636_

Round 1

Reviewer 1 Report

Review of “Simplified Method of Determination of the Sound Speed in Water on the basis of temperature Measurements and Salinity Prediction for Shallow Water Bathymetry” by Artur Makar

The author of this manuscript presents a method to obtain low-cost measurements of shallow water temperature to predict sound speed when salinity variation over the depth is little. This is needed for single beam echo sounder calibration for depth measurement. Though limited to very shallow water regions with small variation in salinity, the results presented show a reasonably high degree of accuracy of predicted sound speeds compared to those using standard formulae. The analyses presented are sound and through, and have the potential to be expanded to more diverse environmental conditions if accurate salinity measurements/predictions are available.

I have two minor comments that will help improve the manuscript:

  • It would be helpful for the readers to understand the degree of errors in various methods as a function of water depth via figures rather than then tables. If the author can present the information in Tables 5-11 in a figure format, with relative errors in sound speed and not absolute values, that would help readers visualize the validity of this approach.

  • It would also help to add an order of magnitude estimate of cost reduction using this method compared to standard methods/CTD probes used for sound speed estimation for a typical bathymetric or biological survey to further emphasize the need for such low-cost methods.

If these changes are made to the manuscript, it is my recommendation that this manuscript is appropriate for publication in Remote Sensing.

Author Response

Dear Reviewer,

I have carefully considered your comments and I found the feedback to be insightful and valuable.  I  thank you  and  the  reviewers for  the  efforts  and  believe  that  I  have  a  much-improved version of my paper. Results are presented in a figure format. Section Discussion has been added. The answers to the reviewers’ comments are in red color in this document.

Artur Makar

Reviewer 2 Report

  1. Taking salinity as a fixed value in sound velocity measurement requires a priori knowledge of salinity in different environments. However, in many environments, salinity varies greatly with depth. How to deal with this problem?
  2. The measurement is only carried out in shallow water environment with little overall change of salinity with depth. How about in deep water environment?

Author Response

(The authors gave the same response as above.)

Reviewer 3 Report

The paper presents results in estimating the speed of sound in water utilising formulas in the literature when a presumed average salinity value is used. The use of an average salinity with depth infers an error in the estimation of the sound speed which in turn results in an erroneous estimation of the depth when employing a single beam echo sounder.

It would be beneficial that the errors in depth be presented as well as the errors in sound speed as this would be the more interesting aspect of this paper. The paper also shows variations during a 24h period of temperature and salinity but does not discuss the variability that is induced on bathymetric measurements. The paper does not provide any new or innovative analysis as it is now. The use of estimated or erroneous values in calculating sound speed is a well-known fact. It would however be beneficial to understand its impact in highly variable environments for the measurement of bathymetry. How could the author attempt to compensate for these effects? Or how could the variability and statistical tools be employed to provide a better estimate of bathymetry and its errors?

I also believe references are lacking, there are methods already seeking to address this problem which are not cited.

Author Response

(The authors gave the same response as above.)

Round 2

Reviewer 3 Report

The reviewer would like to thank the author for addressing the comments.

Author Response

Dear Reviever,

Please find attached file wit the response for your comments.

Artur Makar

Author
